# Correlated Uncertainty for Learning Dense Correspondences from Noisy Labels

**Natalia Neverova, David Novotny, Andrea Vedaldi**
Facebook AI Research
{nneverova, dnovotny, vedaldi}@fb.com

## Abstract

Many machine learning methods depend on human supervision to achieve optimal performance. However, in tasks such as DensePose, where the goal is to establish dense visual correspondences between images, the quality of manual annotations is intrinsically limited. We address this issue by augmenting neural network predictors with the ability to output a distribution over labels, thus explicitly and introspectively capturing the aleatoric uncertainty in the annotations. Compared to previous works, we show that correlated error fields arise naturally in applications such as DensePose and these fields can be modelled by deep networks, leading to a better understanding of the annotation errors. We show that these models, by understanding uncertainty better, can solve the original DensePose task more accurately, thus setting the new state-of-the-art accuracy in this benchmark. Finally, we demonstrate the utility of the uncertainty estimates in fusing the predictions produced by multiple models, resulting in a better and more principled approach to model ensembling which can further improve accuracy.

## 1   Introduction

Deep neural networks achieve state-of-the-art performance in many applications, but at the cost of collecting large quantities of annotated training data. Manual annotations are time consuming and, in some cases, of limited quality. This is particularly true for quantitative labels such as the 3D shape of objects in images or dense correspondence fields between objects. In these cases, one should consider manual labels as a form of weak supervision and design learning algorithm accordingly.

An emerging approach to handle annotation noise is to task the network with predicting the aleatoric uncertainty in the labels. Consider a predictor $\hat{y} = \Phi(x)$ mapping a data point $x$ to an estimate $\hat{y}$ of its label. Given the "ground-truth" label $y$, the standard approach is to minimize a loss of the type $\ell(y, \hat{y})$ so that $\hat{y}$ approaches $y$ as much as possible. However, if the "ground-truth" value $y$ is affected by noise, then naively minimizing this quantity may be undesirable. An alternative approach is to predict instead a *distribution* $p(\hat{y}|x) = \Phi_{\hat{y}}(x)$ over possible values of the annotation $y$. This has several advantages: (1) it can model the distribution of annotation errors specific to each data point $x$ (as not all data points are equally difficult to annotate), (2) it can model the prediction uncertainty (as knowing $x$ may not be sufficient to fully determine $y$), and (3) it allows the model to account for its own limitations (by assessing the difficulty of the prediction task). Under such a model, the point-wise loss is replaced by the negative log-likelihood $\ell(y, \Phi(x)) = -\log p(y|x) = -\log \Phi_y(x)$.

Approaches using these ideas have demonstrated their power in a number of applications. However, most methods have adopted simplistic uncertainty models. In particular, when the goal is to predict a label *vector* $y \in \mathbb{R}^n$, errors have been assumed to be uncorrelated, so that $-\log p(y|x) = -\sum_{i=1}^n \log p(y_i|x)$. However, this is very seldom the case. For example, if $y_i$ is a label associated to a particular pixel $i$ in an image $x$, we can expect annotation and prediction errors to be very strongly correlated for pixels in the same neighborhood.

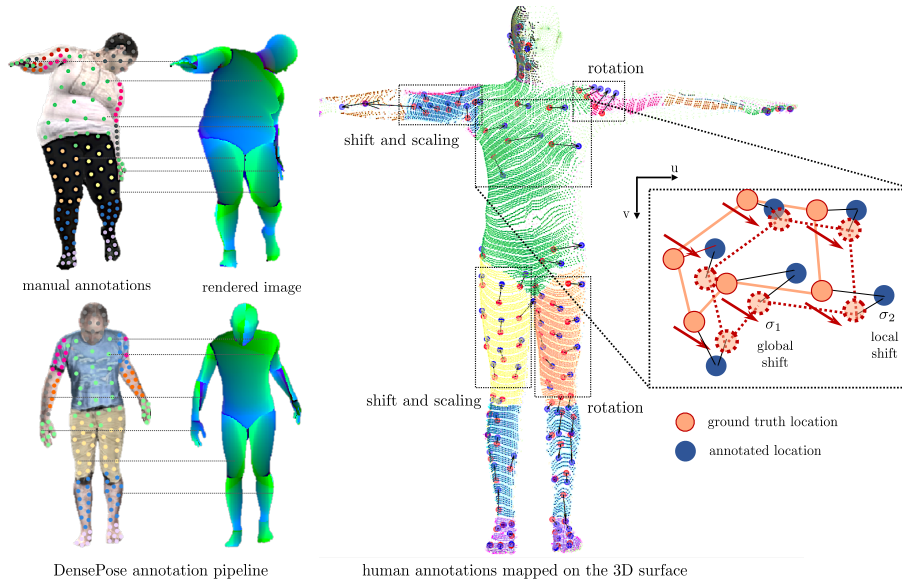

Figure 1: **Systematic errors in manual dense correspondences (DensePose [5]).** The annotators are shown a set of points sampled randomly and uniformly over one of predefined body parts of a person in an image. Their task is to click on corresponding pixels in another image obtained by rendering and therefore providing ground truth correspondences to a canonical 3D model of a human body. Due to self-occlusions and ambiguities, errors made by the annotators tend to be correlated within each given body part and can be partially described by global affine transforms (translation, rotation, scaling) w.r.t true locations. We model the structure of these errors by learning a neural network that estimates a distribution over the correlated annotation field.

In this paper, we investigate richer uncertainty models that can better exploit the data structure. As a running example, we consider the task of dense pose estimation, or *DensePose* (fig. 1): namely, given an image $x$ of a human, the goal is to associate to every pixel $i$ a coordinate chart $(p_i, u_i)$ where $p_i \in \{0, 1, \ldots, P\}$ is a *part label* (where 0 is the background) and $u_i \in [0,1]^2$ is a chart coordinate, specific to part $p_i$ (fig. 1). This is an interesting test case for three reasons: (1) the task consists of pixel-wise, and hence correlated, predictions; (2) the quality of human annotations is uneven and data dependent advantages; (3) there is structure in the data (body parts) that may align with the structure of the annotation errors.

We make three contributions. The first is to apply a standard uncertainty model to this task of learning dense correspondences; this was not done before and alone achieves a significant improvement over strong baselines trained with regression losses.

Our second, and more significant, contribution is to propose more sophisticated uncertainty models. These models allow to predict for each pixel $i$ a *direction* for the error in labelling the chart coordinate vectors. This can for example be used to express different degrees of uncertainty due to foreshortening of a limb. They also allow to express a degree of *correlation* between all vectors that belong to a common region, in this case identified as a human body part. While richer, our models can still be integrated efficiently in deep neural networks for end-to-end training.

Our third contribution is a deeper departure from prior work. Instead of just modelling uncertainty as distribution $-\log p(\hat{y}|x) = \Phi_{\hat{y}}(x)$ conditioned on the input data $x$, we consider the possibility of conditioning the uncertainty on *the annotation $y$ directly*. For example, in the DensePose task, the annotation $y$ can be used to predict image regions where uncertainty is likely to be higher even before observing the image $x$.

## 2    Related work

Uncertainty in machine learning is usually decomposed into three types [9]: approximation, due to the fact that the model is not sufficiently expressive to model the data-to-label association, aleatoric,

due to the intrinsic stochastic nature of the association, and epistemic, due to the model's limited knowledge about this association, which prevents it form determining it uniquely.

Uncertainty in deep neural network has been modelled using approximate Bayesian approaches [6, 1, 7, 8] using dropout as a way to generate the necessary samples [4, 7]. Ensembling [3], which combines multiple models, has been explored in [15, 10, 16]. The recent method of [18] proposes a frequentist method that can estimate both aleatoric and epistemic uncertainty.

The work of [7, 14] is probably the most related to ours. They also model approximation and aleatoric uncertainty by configuring a deep neural network to produce an explicit estimate of the latter, in the form of a parametric posterior distribution. In this paper, we build a similar model, but apply it to a dense, structure image labelling tasks. We thus extend the model to express structured uncertainty, where errors are highly correlated in a way which depends on the input image and annotation.

We apply our approach to the DensePose problem, originally introduced in [5]. We not only show that we can accurately model uncertainty in the annotation process, but also learn better overall DensePose regressor, outperforming the current state-of-the-art results of [12], from which we borrow our experimental setup.

## 3    Method

We consider the problem of predicting, given a data point $x$, a label vector $y \in \mathbb{R}^{dn}$ formed by $n$ subvectors $y_i \in \mathbb{R}^d, i = 1, \dots, n$ of dimension $d$. In our test application, namely DensePose, these subvectors are the chart coordinates $y_i = u_i \in [0,1]^2$ that associate to pixels $i$ of image $x$ a particular point on the human body. However, there are many other problems, including colorization, depth prediction and inpainting, that can be modelled in a similar manner.

We denote by $\delta = \hat{y} - y$ the error between the *predicted* value $\hat{y}$ of the label and the annotated value $y$. In order to model uncertainty in the system, we train a predictive model $\Phi(x)$ that outputs not only the point estimate $\hat{y} \approx y$, but also a distribution $p(y|x)$ over possible values. For simplicity, we express the latter as

$$p(y|x) = q(\hat{y} - y|x) \tag{1}$$

where $q(\delta|x)$ is an unbiased distribution of the residual (i.e. $E[q(\delta|x)] = 0$ and $\mathrm{argmax}_\delta\, q(\delta|x) = 0$). Hence, the output of the neural network is a pair $(\hat{y}, q) = \Phi(x)$ comprising a point estimate $\hat{y}$ and the distribution of the residual $q$. This model can be trained by optimizing the negative log-likelihood $\ell(y, \Phi(x)) = -\log q(y - \hat{y})$.

Next, we discuss possible variants of the model with different complexity and expressive power.

### 3.1    Elementary uncertainty model

In the simplest case, we let $y \in \mathbb{R}^n$ and assume that subvectors $y_i$ are single scalars. The uncertainty model then is given by $q(\delta|x) = \prod_{i=1}^{n} q(\delta_i|x)$ which amounts to assuming that the residuals are statistically independent. The simplest choice for $q(\delta_i|x)$ is a Gaussian $\mathcal{N}(0, \sigma_i^2)$ with standard deviation $\sigma_i^2$. Hence, the neural network $(\hat{y}, \sigma) = \Phi(x)$ outputs for each pixel $i$ the prediction $\hat{y}_i$ as well as an estimate $\sigma_i^2$ of the prediction uncertainty. In this case, the training loss expands as:

$$\ell(y, \Phi(x)) = \frac{n}{2}\log 2\pi + \frac{1}{2}\sum_{i=1}^{n}\left(\log \sigma_i^2 + \frac{(\hat{y}_i - y)^2}{\sigma_i^2}\right) \tag{2}$$

Note that minimum of the r.h.s. w.r.t. $\sigma_i^2$ is obtained by setting $\sigma_2 = |\hat{y}_i - y|$. Hence, the predictor $\Phi$ will try to output a value of $\sigma_i$ which is equal to the error actually incurred at that pixel; crucially, however, the model $\Phi$ *cannot* measure this error directly as it only receives the data point $x$ (and not the annotation $y$) as input. Hence, the model is encouraged to perform *introspection*.

### 3.2    Higher-order uncertainty models

The model of section 3.1 is simplistic as it assumes that errors are statistically independent, which is seldom the case in applications. In order to address this limitation, we assume that residuals are instead generated by the model

$$\delta_i = \epsilon + \eta_i + \xi_i w_i \tag{3}$$

where $\epsilon \sim \mathcal{N}(0, \sigma_1^2 I_d)$ is an overall isotropic offset, $\eta_i \sim \mathcal{N}(0, \sigma_{2i}^2 I_d)$, is a subvector specific isotropic offset and $\xi_i \sim \mathcal{N}(0, \sigma_{3i}^2)$ is a subvector specific directional offset along the unit vector $w_i$. Here $I_d$ denotes the $d \times d$ identity matrix, where $d$ is the dimensionality of the subvectors $y_i$ ($d = 2$ in the DensePose application).

This model extends (2) in several ways. First, the term $\epsilon$ indicates that errors are overall correlated. For instance, in DensePose it is likely that all points annotated for a given human body part would be affected by a similar annotation shift compared to the "correct" annotation. This is because humans are better at relative rather than absolute judgments when it comes to establishing ambiguous visual correspondences. Second, the term $\eta_i$ expresses local isotropic uncertainty, similar to (2). Third, the term $\xi_i w_i$ express local *directional* uncertainty. This can be used to capture any expected directionally in the error. For instance, in DensePose we expect errors to be larger in the direction of visual foreshortening of a limb.

Next, we calculate the negative log-likelihood $-\log q(\delta|x)$ under this model in order to evaluate and learn it. The collection of residuals $\delta$ is a Gaussian vector with co-variance matrix

$$\Sigma = JJ^\top + \mathrm{diag}(\Sigma_1, \ldots, \Sigma_n), \quad \Sigma_i = \sigma_{2i}^2 I_d + \sigma_{3i}^2 w_i w_i^\top, \tag{4}$$

where $J = \sigma_1 \cdot [I_d \quad \cdots \quad I_d]^\top \in \mathbb{R}^{dn \times d}$.

Some algebra shows that the determinant of the covariance matrix and the concentration matrix are given by

$$\det \Sigma = \det K \cdot \prod_{i=1}^n \det \Sigma_i, \quad C = \Sigma^{-1} = \bar{C} - \bar{C} J K^{-1} J^\top \bar{C}, \quad K = I_d + \sigma_1^2 \cdot \sum_{i=1}^n \Sigma_i^{-1}. \tag{5}$$

Here $\bar{C} = \mathrm{diag}(C_1, \ldots, C_n)$ is the block-diagonal matrix containing the subvector-specific concentration matrices $C_i = \Sigma_i^{-1}$. We can expand this further by noting that:

$$C_i = \frac{1}{\sigma_{2i}^2} \cdot \Pi_i, \quad \Pi_i = I_d - \rho_i w_i w_i^\top, \quad \det \Sigma_i = \frac{\sigma_{2i}^{2d}}{1 - \rho_i}, \quad \rho_i = \frac{\sigma_{3i}^2}{\sigma_{2i}^2 + \sigma_{3i}^2} \tag{6}$$

where $\Pi_i$ can be interpreted as a projection operator and $\rho_i$ as a correlation coefficient. Then:[1]

$$-\log q(\delta|x) = \frac{nd}{2} \log 2\pi + \frac{1}{2} \sum_{i=1}^n \left( \log \frac{\sigma_{2i}^{2d}}{1 - \rho_i} + \frac{\delta_i^\top \Pi_i \delta_i}{\sigma_{2i}^2} \right)$$

$$+ \frac{1}{2} \log \det K - \frac{\sigma_1^2}{2} \left( \sum_{i=1}^n \frac{\Pi_i \delta_i}{\sigma_{2i}^2} \right)^\top K^{-1} \left( \sum_{i=1}^n \frac{\Pi_i \delta_i}{\sigma_{2i}^2} \right), \quad K = I_d + \sigma_1^2 \cdot \sum_{i=1}^n \frac{\Pi_i}{\sigma_{2i}^2}. \tag{8}$$

**Spatially-independent model.** Model (8) correlates the errors of all subvectors. We can loose this condition by setting $\sigma_1 = 0$. In this case $K = I_d$ and $J = 0$ and the model reduces to:

$$-\log q(\delta|x) = \frac{nd}{2} \log 2\pi + \frac{1}{2} \sum_{i=1}^n \left( \log \frac{\sigma_{2i}^{2d}}{1 - \rho_i} + \frac{\delta_i^\top \Pi_i \delta_i}{\sigma_{2i}^2} \right). \tag{9}$$

$$\frac{\sigma_{2i}^{2d}}{1 - \rho_i} = \sigma_{2i}^{2(d-1)}(\sigma_{2i}^2 + \|r_i\|^2), \qquad \rho_i = \frac{\|r_i\|^2}{\sigma_{2i}^2 + \|r_i\|^2}, \qquad \Pi_i = I_d - \frac{r_i r_i^\top}{\sigma_{2i}^2 + \|r_i\|^2}. \tag{7}$$

Furthermore, the negative sign in eq. (8) can lead to instabilities. We thus rewrite the equation as the sum of squares:

$$\frac{\delta_i^\top \Pi_i \delta_i}{\sigma_{2i}^2} - \frac{\sigma_1^2}{2} \left( \sum_{i=1}^n \frac{\Pi_i \delta_i}{\sigma_{2i}^2} \right)^\top K^{-1} \left( \sum_{i=1}^n \frac{\Pi_i \delta_i}{\sigma_{2i}^2} \right) = \frac{1}{2} \sum_{i=1}^n (\delta_i - \mu)^\top \frac{\Pi_i}{\sigma_{2i}^2} (\delta_i - \mu) + \frac{\mu^\top \mu}{2\sigma_1^2}$$

where we have introduced the 'mean' vector:

$$\mu = \sigma_1^2 K^{-1} \sum_{i=1}^n \frac{\Pi_i}{\sigma_{2i}^2} \delta_i.$$

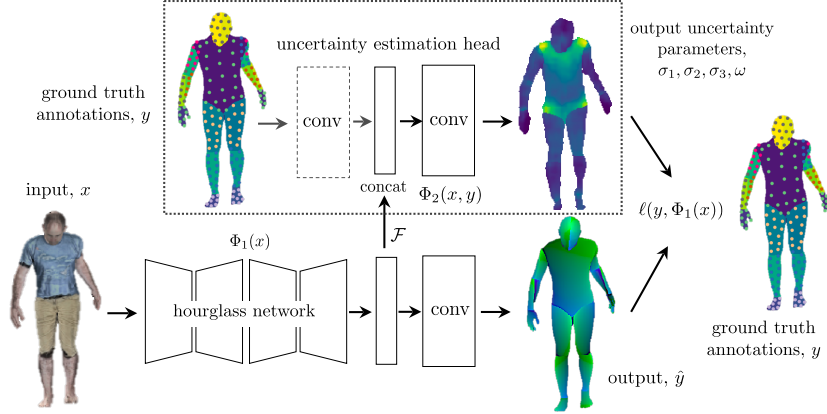

Figure 2: **Uncertainty-aware training pipeline.** We extend a standard predictor based on a Hourglass architecture [13, 12] with an additional *uncertainty head* to estimate uncertainty parameters.

This requires the model to estimate for each pixel the value of the variance $\sigma_{2i}^2$ as well as of the directional correlation parameters $r_i \in \mathbb{R}^2$.

**Fully-independent model.** The elementary model corresponding to eq. (2) is obtained by further setting parameter $r_i = 0$ in eq. (9).

### 3.3 Label-conditioned uncertainty

Next, we consider modifying the approach so that uncertainty is predicted not only from the input data $x$, but also based on the label $y$ itself.

On a first glance, this may look as simple as adding the argument $y$ to the predictor $\Phi(x)$ to obtain a new predictor $\Phi(x, y)$. This is however nonsensical as $(\hat{y}, q) = \Phi(x, y)$ is tasked with producing an estimate of $\hat{y}$ itself, so this would immediately lead to a degenerate solution.

Instead, we consider two separate networks. The first, $\hat{y} = \Phi_1(x)$, is tasked with predicting only the label $y$. The second, $q = \Phi_2(x, y)$, is tasked with predicting only the uncertainty distribution $q$. Without any constraint, this scheme still does not work as the distribution $q$ can shift the prediction $\hat{y}$ arbitrarily. This is prevented by the fact that $E[q(\delta)] = 0$; in fact, in practice we require $q(y)$ to be a simple uni-modal distribution. In this manner, $q$ can effectively only predict the data uncertainty, but $\hat{y}$ must still try to predict the label correctly in order to minimize the log-likelihood loss.

### 3.4 Introspective ensemble

Assume that we have densities $q_k(\delta|x)$, $k = 1, \ldots, K$ generated from an ensemble of $K$ models and let $\hat{y}^{(k)}$ be the corresponding label estimates (we use the superscript to indicate that we index different estimates instead of different components of a single vector estimate). We can fuse the estimates by finding $y = \operatorname{argmax}_y \sum_{k=1}^{K} \log q_k(y - \hat{y}^{(k)}|x) = \operatorname{argmax}_y \sum_{k=1}^{K} (y - \hat{y}^{(k)})^\top C^{(k)}(y - \hat{y}^{(k)})$. The maximizer is then $y = (\sum_{k=1}^{K} C^{(k)})^{-1} \sum_{k=1}^{K} C^{(k)} \hat{y}^{(k)}$. Note that, while the section above gives us the inverse of the concentration matrices of each model, in case of the probabilistic model of the highest order eq. (8) we require the inverse of the sum, which must be obtained numerically. In time-constrained applications, such as real time, rather than solving such a large system of equations, we can utilize conjugate gradient descend to obtain an approximate solution starting from an initial guess (which can be obtained as the average of the individual models' predictions). For the simpler cases of ensembles of spatially-independent models and fully-independent models, the fused estimates can be computed in closed form. For the spatially independent model (eq. (9)), it follows that the fused $uv$ predictions $y_i^{spa}$ at position $i$ are defined as $y_i^{spa} = (\sum_{k=1}^{K} C_i^{(k)})^{-1}(\sum_{k=1}^{K} C_i^{(k)} \hat{y}_i^k)$, which now only requires to invert a small 2x2 matrix formed by accumulating $C_i^{(k)}$ above each pixel. In case of

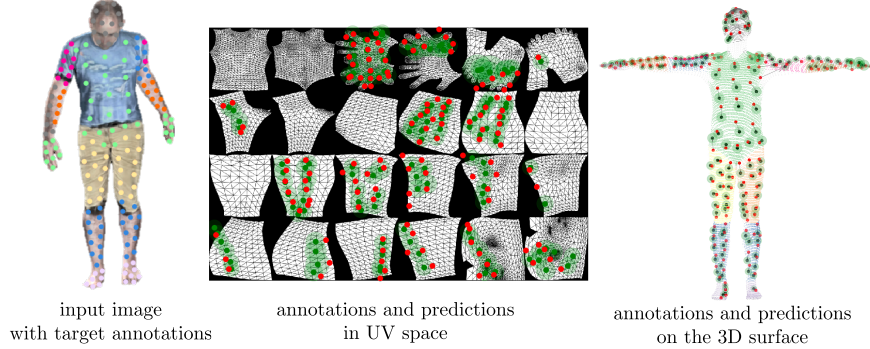

<div align="center">

input image                  annotations and predictions           annotations and predictions
with target annotations                  in UV space                     on the 3D surface

</div>

Figure 3: **Example of predictions produced by our model.** Ground truth locations (in red) and predicted locations (in green) are shown together with learned isotropic offsets described by $\sigma_2$.

the fully independent model with isotropic covariance matrices (eq. (2)), the ensembled prediction $y_i^{iso} = \sum_{k=1}^{K} \frac{\hat{\sigma}_i^{-2(k)}}{\sum_{k=1}^{K} \hat{\sigma}_i^{-2(k)}} \hat{y}_i^{(k)}$ is a mere weighted sum of $\hat{y}_i^{(k)}$.

## 4 Application to DensePose

In this section we show in more detail how the ideas explained above can be applied to the DensePose problem [5]. In this work, we adapt the DensePose setting of [12], where the input is an image $x \in \mathbb{R}^{3 \times H \times W}$ tightly containing a person (DensePose can also be applied to full images in combination with an object detector, but we are not concerned with that here). DensePose then trains a network $\Phi$ to predict a label $y \in \mathbb{R}^{C \times H \times W}$ where the $C = 3 \cdot P$ channels of vector $y_i$ for pixel $i$ comprise a $P$-dimensional indicator vector for the part that contains pixel $i$ (e.g. left forearm) and the 2D location in the chart of each part, accounting for $2P$ dimensions. Note that only one of the $2P$ predicted locationa is used at pixel $i$, as indicated by the part selector, but all of them are still computed.

We extend the basic architecture with an additional *uncertainty head* that estimates the prediction confidences (see fig. 2). Depending on which specific model is implemented, the output dimensionality of this branch may differ. However, each variant amounts to predicting a certain number of additional channels per part, estimating part-specific uncertainty values, and can be generally expressed as $N_u \times P$, where $N_u$ is the number of uncertainty parameters. Note that depending on the application, at test time the uncertainty head may be either utilized to get confidence estimates, or ignored. In the latter case, the uncertainty aware training results in a boost of model performance at no extra computational cost during inference.

All uncertainty parameters are predicted by applying a set of two convolutional blocks to an intermediate feature level $\mathcal{F}$, produced by the main network $\Phi_1$. As mentioned in section 3.3, in addition, we explore a variant of the model where the sparse ground truth annotations are passed directly to the uncertainty head as an additional input. The annotated points are first mapped onto the image space, preprocessed by a set of partial convolutional layers [11] and then concatenated with features $\mathcal{F}$. This process is illustrated in fig. 2. An example of model predictions is shown in fig. 3.

## 5 Experiments

**Datasets.** To gain deeper insights on the nature of human annotator errors on the dense labeling task, we first analyzed DensePose annotations obtained on a set of 88 synthetic images, where ground truth UV mapping is known by design (analogously to Section 2.2 of [5]). These images were rendered using the SMPL body model [2] and the rendering pipeline of [19].

We have empirically observed that, by taking all annotated points covering one body part in one image and applying a simple global affine transformation to them (such as translation or scaling), the mean error over the whole image set can be reduced by half. This confirms our hypothesis of existing strong correlation between individual errors.

| Model | uv-loss | 1 cm | 2 cm | 3 cm | 5 cm | 10 cm | 20 cm |
|---|---|---|---|---|---|---|---|
| DensePose-RCNN (R50) [5] | MSE | 5.21 | 18.17 | 31.01 | 51.16 | 68.21 | 78.37 |
| | full (ours) | **5.67** | **18.67** | **32.70** | **53.14** | **71.25** | **80.47** |
| HRNetV2-W48 [17] | MSE | 4.31 | 15.19 | 27.14 | 47.07 | 69.76 | 78.66 |
| | full (ours) | **5.70** | **18.81** | **31.88** | **52.20** | **74.21** | **82.12** |
| HG, 1 stack (Slim DensePose [12]) | MSE | 4.31 | 15.62 | 28.30 | 49.92 | 74.15 | **83.01** |
| | full (ours) | **5.34** | **18.23** | **31.51** | **52.40** | **74.69** | 82.94 |
| HG, 2 stacks (Slim DensePose [12]) | MSE | 4.44 | 16.21 | 29.64 | 52.23 | 76.50 | **85.99** |
| | full (ours) | **5.99** | **19.97** | **34.16** | **55.68** | **77.76** | 85.58 |
| HG, 8 stacks (Slim DensePose [12]) | MSE | 6.04 | 20.25 | 35.10 | 56.04 | 79.63 | 87.55 |
| | full (ours) | **6.41** | **20.98** | **35.17** | **56.48** | **80.02** | **87.96** |

Table 1: **Performance of uncertainty-based models on the DensePose-COCO dataset [5]**. Our models significantly outperform the baseline variants, with no extra computational cost at inference (when uncertainty estimates are not required by application).

| Data | SMPL renderings (synthetic) | | | DensePose-COCO (real) | |
|---|---|---|---|---|---|
| | *MAP* | gt-real | gt-human | *MAP* | gt-human |
| simple | *(2.9785)* | 1.0816 | 1.1716 | *(3.0797)* | 1.3159 |
| simple-2D | *(2.5748)* | 1.4246 | 1.4825 | *(2.5892)* | 1.3651 |
| iid | *(3.2285)* | 1.7026 | 1.8937 | *(3.2038)* | 1.4383 |
| full | *(3.0683)* | **2.3448** | **2.3574** | *(2.9847)* | **2.14057** |

Table 2: **Negative log-likelihood of human annotations under different models with uncertainty.** More advanced models show monotonic increase in this metric w.r.t the ground truth locations. gt-human stands for human annotations on synthetic data, gt-real for the ground truth UV maps. simple-2D: assumes independent (but not isotropicaly nor identically distributed) per-pixel errors.

The majority of experiments in this work were conducted on the DensePose-COCO dataset [5], containing 48k densely annotated people in the training set and 2.3k in the validation set. We follow the *single person* protocol for this task and use ground truth annotations for bounding boxes to crop images around each person.

**Metrics.** For evaluation, we adapt a standard per-pixel metric used in [5] and [12] and report the percentage of points predicted with an error lower than a set of predefined thresholds, where the error is expressed in geodesic distances measured on the surface of the 3D model. Since, in this work, we focus specifically on the UV-regression part of the DensePose task while keeping the segmentation pipeline standard, we additionally report performance w.r.t. stricter geodesic thresholds and in two settings when the segmentation is either predicted by the same network or assumed to be perfect and correspond to the ground truth at test time.

**Implementation details.** The architecture of the main DensePose predictor $\Phi_1$ is based on the Hourglass network [13] adapted to this task by [12]. We benchmark performance on 1, 2 and 8 stacks, but conduct most of the ablation studies on a 1-stack network for speed. All networks are trained for 300 epochs with SGD, batch size 16 and learning rate of 0.1 decreasing by a factor of 10 after 180 and 270 epochs. Input images are normalized to the resolution of $256 \times 256$.

**Uncertainty models.** In our experiments, we analyze several modifications of networks with uncertainty heads, which we denote as follows: MSE stands for the uncertainty free baseline of [12] trained with the MSE regression loss; simple corresponds to the elementary uncertainty model given by 2; simple-2D is a variant of (2) with two distinct $\sigma_u$ and $\sigma_v$ learned separately for u- and v-dimensions; iid denotes the spatially-independent model of (9) and full stands for the complete model given defined by (8).

As shown in Table 3, introducing uncertainty into the DensePose training brings significant gains in performance over the considered baseline. Our 2-stack architecture significantly outperforms

| Model | UV only (GT body parsing) | | | | | overall performance | | | | |
|---|---|---|---|---|---|---|---|---|---|---|
| | 1 cm | 2 cm | 3 cm | 5 cm | 10 cm | 1 cm | 2 cm | 3 cm | 5 cm | 10 cm |
| MSE | 5.57 | 19.92 | 35.74 | 61.75 | 89.53 | 4.31 | 15.62 | 28.30 | 49.92 | 74.15 |
| simple | 6.71 | 22.80 | 38.97 | 64.30 | 90.04 | 5.27 | 17.96 | 31.08 | 52.04 | 74.39 |
| simple-2D | 7.02 | 23.40 | 39.76 | 64.89 | 90.25 | 5.54 | 18.53 | 31.80 | 52.67 | 74.98 |
| iid | 6.95 | 23.39 | 39.70 | 64.95 | 90.25 | 5.44 | 18.46 | 31.70 | 52.66 | 74.84 |
| full | 6.78 | 23.08 | 39.42 | 64.62 | 90.14 | 5.34 | 18.23 | 31.51 | 52.40 | 74.69 |

Table 3: **Ablation on uncertainty terms.** The left part of the table reports upper bound results in assumption of perfect body parsing at test time.

| Model | UV only (GT body parsing) | | | | | overall performance | | | | |
|---|---|---|---|---|---|---|---|---|---|---|
| | 1 cm | 2 cm | 3 cm | 5 cm | 10 cm | 1 cm | 2 cm | 3 cm | 5 cm | 10 cm |
| full | 6.78 | 23.08 | 39.42 | 64.62 | 90.14 | 5.34 | 18.23 | 31.51 | 52.40 | 74.69 |
| +gt (train) | **7.18** | **23.84** | **40.33** | **65.36** | **90.38** | **5.60** | **18.68** | **31.97** | 52.57 | 74.28 |

Table 4: **Label-conditioned uncertainty.** Exploiting ground truth labels as an additional direct cue has shown to accelerate learning and results in higher performance.

| Model | Best model | | | Average | | | Ours | | |
|---|---|---|---|---|---|---|---|---|---|
| | 5 cm | 10 cm | 20 cm | 5 cm | 10 cm | 20 cm | 5 cm | 10 cm | 20 cm |
| MSE | 49.92 | 74.15 | 83.01 | 50.49 | 75.09 | 83.82 | – | – | – |
| simple | 52.04 | 74.39 | 82.74 | 54.26 | 75.49 | 82.84 | **54.46** | **75.55** | **82.86** |
| iid | 52.66 | 74.84 | 83.01 | 54.15 | 75.29 | 82.56 | **54.55** | **75.59** | **82.77** |

Table 5: **Introspective ensemble**. We collect a number of models with identical architectures but trained with different hyperparameters (weights on the UV-term: 0.1, 0.2 and 0.5). More diverse ensembles are expected to deliver higher gains in performance in all settings.

much more powerful 8-stack models, especially when measured on tighter geodesic thresholds. This provides an additional evidence for the hypothesis that uncertainty-based training facilitates learning with noisy annotations and allows the model to decrease the associated jitter in predictions.

The ablation on different variants of the uncertainty models is given in Table 3. In terms of accuracy of UV-predictions, the more advanced models (iid and full) perform on par with their simpler counterparts (simple and simple-2D) (note that test complexity of the main predictor is identical for all models). Their advantage however becomes apparent when looking at the log-likelihood of the ground truth labels evaluated by each of the models (see Table 2). In this setting, full model clearly provides more meaningful representation of the learned distribution, which is no doubt critical for numerous downstream tasks.

**Label-conditioned uncertainty.** Following the discussion of Section 3.3, we ablated the effect of using ground truth annotations as a direct cue for learning uncertainty. Table 4 shows immediate benefits of doing so in terms of the target UV-metrics, but we also observed significant increase of log-likelihoods of the true labels evaluated by the gt-based model. Note that no ground truth information is required by the model at test time, as long as uncertainty estimates are not utilized.

**Introspective ensembles.** Finally, we benchmark the performance of the proposed ensembling techniques for several variants of the models with uncertainty. Exploiting uncertainty parameters for finding the right balance consistently outperforms the averaging late fusion baseline in all tested scenarios over a range of models (see Table 5).

## 6 Conclusions

In this paper we have investigated the use of introspective uncertainty prediction models to improve the robustness and expressiveness of models for dense structured label prediction. We have introduced

a method to estimate, using a convolutional neural network, an uncertainty model which potentially correlates the errors of all the pixels in an image. We have applied these ideas to the DensePose tasks, showing how these approaches can result in significant performance improvements compared to the current state-of-the-art. Since the structure of the regressor is unchanged compared to the latter approaches, these improvements are solely imputable to the models' better understanding of the uncertainty in the data. This is particularly beneficial for problems, such as DensePose, where the quality of manual labels is intrinsically limited.

## Footnotes

[1] In practice, it is easier for a network to predict $r_i = \sigma_{3i} w_i \in \mathbb{R}^2$ instead of $\rho_i$ and $w_i$ separately, so we use the parametrization

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
