[Reviews · NeurIPS 2019]

Reviewer 1



Originality & Clarity & significance The technique presented in the paper is quite novel. The paper is generally well written and easy to read. The method will potentially improve a lot of existing algorithms. Quality I am satisfied with the methodology sections. However, I have quite a few questions regarding the experimental part: 1. The experimental settings are not clear to me. For the label-conditioned branch, which I believe as one of the most important parts (e.g., the problem is raised in l-101?), there seems only one experiment in Table 4 is performed. Is it true? Does it mean all the other evaluations and ablation studies are conducted with a single branch, rather than the full network shown in Figure 2? 2. What is the actual difference between simple-2D and full? 3. I found the evaluation choices are random. E.g., (a) Table 1 shows only the performance of full model (this is fine), but there is not full model performance for 8 stacks. (b) There is a +gt only for the full model in Table 4. (c). Table 5 does not report the simple-2D and full, and the 1 cm and 2 cm. It is better to provide either consistent or full results.

Reviewer 2



The paper considers the task of dense pose estimation, which can be decomposed into finding a semantic segmentation of body parts complemented with the regression of u-v coordinates for each pixel, relative to each body part. The authors explore several error models which all result in the addition of output heads to an existing dense pose network. The simplest one is a local error model, the most complex model (the `full’ model) considered, is one that adds in a global error plus local errors that are independent for the u-v coordinates. The output heads are trained by maximum likelihood for the regression model of the u-v coordinates. The full model performs as well as the simpler error models but results in a higher neg. log-likelihood and is therefore presented as superior. Two other cases are considered: 1) training a model whose output heads for the error estimation are additionally conditioned on the ground truth and 2) a way of combining independently trained models by using their uncertainty estimates. It is unclear why the authors do not present the results of related work, which makes it somewhat difficult to assess their models’ performance, but of course still allows to compare their relative performance. Although the work ablates the model adaptations it considers, it does not seem to discuss the results and implications very well. For instance the simpler models (simple-2D) perform slightly better than the full error model, which however in turn receives a better neg. log-likelihood. Why is that? The model ensembling using uncertainty predictions only performs marginally better than a simple model average, there is no discussion or significance assessment. The model whose uncertainty heads are conditioned on the ground truth during training perform better at test time (w/o access to the ground truth). It is unclear why this conditioning helps training of the core prediction network beyond using the regression loss alone. There is no discussion of that. This model could also not be used to assess the model’s uncertainty at test time, also prohibiting an ensemble of this kind of model. The term `introspection ensemble’ seems a bit far-fetched, more accurately it is an uncertainty-weighted ensemble. Being concerned with highly structured aleatoric uncertainty and modeling uncertainty in the annotation process, there is missing related work, e.g. the Probabilistic U-Net which models aleatoric uncertainty for semantic annotations. Also, the error model is concerned with modelling the error of subvectors in u-v-space but not between the part label predictions, i.e. the inter subvector covariance and thus the semantic segmentation, why was there no attempt to incorporate this? Another current limitation of the approach is that the error model does not consider the correlation of errors specific to regions, e.g. individual body parts. Instead they are either local and/or global, despite the discussed observation that the error of individual body parts may be strongly correlated (lines 193 - 196). The discussion of why learning with an uncertainty model helps training and final performance seems insufficient. It is stated that a `model’s better understanding of the uncertainty of the data’ helps. The reason that as to why it helps is presumably the loss attenuation, that allows the model to down-weight the loss of difficult and thus likely ambiguous / noisy examples or pixels. There is quite a bit of literature on that, which could be part of a discussion. Lastly some typos need fixing: `ensemlbing’, `cosntrained’, `gradient descend’, `locationS’.

Reviewer 3



The paper presents a mathematical framework that estimates uncertainty of dense correspondences from noisy labels. The underlying framework models the distribution of residuals obtained by comparing a target vector encoding the correspondences and other properties and its prediction. The underlying model considers three sources of noise: a general noise affecting all dimensions of residuals, noise affecting the dimensions encoding the association of pixels to a human body part, and directional noise modeling directional errors. All three sources of noise are modeled with Gaussian distributions. I think the framework is solid given the assumption of Gaussian distribution. However, I have a major and a few minor concerns. 1. My major concern is that the submission is missing other baselines on human dense correspondences in the experiments. The only baseline is based on [13]. However, there exist other recent methods (e.g., Dense Human Body Correspondences Using Convolutional Networks by Wei et al.). The experiments indeed show improvements over [13] but it is unclear if the paper is advancing the state of the art by including the uncertainty model. In sum, I think the paper should include other CNNs for dense correspondences and include the proposed framework to it and show if the benefit is consistent across other models. 2. A minor concern has to do with the lack of justification of using Gaussian distributions to model the uncertainties. Although I understand that a Gaussian distribution typically simplifies the math, I don't understand why a Gaussian distribution is a good model for the residuals dealt with this problem. 3. A third minor concerns are a few typos and grammatical errors. a) In line 67: "The recent method of [17] proposes a frequentist *methods* ..." -> "The recent method of [17] proposes a frequentist *method* ..." b) In line 85: "In order do [...]" -> "In order to [...]" c) In line 88: I think you mean E_q[\delta] = 0 (i.e., the expected residual is zero); or am I missing something? d) In line 99: When defining \sigma_2, what is u? It was never defined. e) In line 105: "In oder to [...]" -> "In order to [...]" f) In line 126: I think Eq. 7 is just floating around w/o any introduction first. I would suggest gently introducing the Equation in line 126. g) Eq. (9) is missing a period. h) In parts of the text, the Equations are note properly references (e.g., line 216 and line 218). Please clearly refer to them as Eq. (X) in the text. Post Rebuttal: The rebuttal addressed my major concern which was the lack of comparisons with other baselines. The new results clearly show a marginal advantage, and they should be included in the final paper. However, I would encourage the authors to discuss why a Gaussian distribution is good for modeling the errors. Just stating that it is good because it is unimodal is not a good argument. There are many other unimodal distributions (e.g., Laplacian distribution), and I think it would be nice if the paper shows a histogram of the errors and the fit of a Gaussian to them.

[Author Response · NeurIPS 2019]

| Model | uv-loss | 1 cm | 2 cm | 3 cm | 5 cm | 10 cm | 20 cm |
|---|---|---|---|---|---|---|---|
| DensePose-RCNN (R50) [5] | MSE | 5.21 | 18.17 | 31.01 | 51.16 | 68.21 | 78.37 |
| | full (ours) | **5.67** | **18.67** | **32.70** | **53.14** | **71.25** | **80.47** |
| HRNetV2-W48 [*] | MSE | 4.31 | 15.19 | 27.14 | 47.07 | 69.76 | 78.66 |
| | full (ours) | **5.70** | **18.81** | **31.88** | **52.20** | **74.21** | **82.12** |
| HG, 1 stack (Slim DensePose [12]) | MSE | 4.31 | 15.62 | 28.30 | 49.92 | 74.15 | **83.01** |
| | full (ours) | **5.34** | **18.23** | **31.51** | **52.40** | **74.69** | 82.94 |
| HG, 8 stacks (Slim DensePose [12]) | MSE | 6.04 | 20.25 | 35.10 | 56.04 | 79.63 | 87.55 |
| | full (ours) | **6.41** | **20.98** | **35.17** | **56.48** | **80.02** | **87.96** |

Table 1: **Performance of uncertainty-based models on the DensePose-COCO dataset [5].** [*] Sun et al. High-Resolution Representations for Labeling Pixels and Regions. arXiv:1904.04514v1, 2019.

**1: R1: The label-conditioned branch . . . seems [to be] only in Tab. 4. R2: The model whose uncertainty heads are conditioned on the ground truth during training performs better at test time.** There are two reasons for modelling uncertainty: (i) to better understand systematic annotation errors at training time, which leads to more robust training and better point-wise prediction accuracy at test time and (ii) to be able to predict uncertainty at test time, regardless of whether this also results in better point-wise prediction.

Effect (i) was observed in several papers (e.g. [14]) and is mostly due to the ability of the model to detect and discount annotation errors and very hard examples.

Conditioning on the ground-truth part labels is useful for (i) but not for (ii) (because part labels are not available at test time). Since our goal is to *also* achieve (i), we focus on the conditioned models for (ii) in Tab. 4 and use the non-conditioned models in the other experiments. We have now conducted additional experiments for Tab. 4 using conditioned variants of the simple and iid models (in addition to the full as already in the table) and observed consistent gains (0.4-0.6pp @5cm, UV only).

**2: R1: Difference between simple-2D and full.** simple-2D: assumes per-pixel error vectors to be independent (but not isotropicaly nor identically distributed); full: captures the correlation between per-pixel errors.

**3: R1: I found the evaluation choices are random.** As requested, we have filled some gaps in the tables: For Tab. 1 in the paper, the HG-8stack performance of the full model (see Tab. 1 above). For Tab. 4: the performance of all models with uncertainty (see answer 1). For Tab. 5: the performance with tight thresholds with ensembling (similar gains 0.2-0.4pp@2cm, UV only, observed everywhere).

**4: R1: Simple-2D... best... in Table 3 with tight thresholds? R2: Simple-2D perform slightly better than the full error model, which however in turn receives a better neg. log-likelihood. Why?** In practice, all our models that use uncertainty improve the *average* per-pixel prediction errors (PPE) by a similar amount. However, the full model *also* captures the error distribution better (because the errors between different pixels are highly correlated), which is reflected in the higher likelihood but not necessarily reflected in a lower average PPE. This is because average PPE is merely a marginal statistic which ignores the correlations predicted by our models.

**5: R1: Is the log-likelihood directly comparable?** Yes, all models define a distribution on the same variables.

**6: R1: Is the uncertainty not fully correlated to the dense pose performance?** See answer 4.

**7: R2: do not present the results of related work. R3: The only baseline is based on [13].** We report & outperform the Thrifty DensePose baseline of [12], which is near state-of-the-art for the problem of dense pose recognition (see also table at the top) (Parsing R-CNN is slightly better, but their models are unavailable). In Tab. 1 above, we also compare to the original DensePose-RCNN [5] and additionally report performance using the HRNet architecture (state-of-the-art in pose estimation and semantic segmentation) applied to the dense pose estimation task. In all cases, our models show consistent gains over the whole range of thresholds.

**8: R2: Significance of ensembling.** Considering that predictions of the ensemble do not significantly differ (as noted in capt. of Tab. 5), which is a necessary condition for better performance, we find the improvement satisfactory.

**9: R2: Related... Probabilistic U-Net.** Will add & discuss.

**10: R2: [does not model] the error between the part label predictions... nor... correlation of errors specific to regions.** Model (3) *does* capture the correlations of error vectors within each region via the error term $\epsilon$. Note, in particular, that this term is part-specific, not global. Part-labelling errors are also important, but accounting for them would require a dramatically more complex model due to the resulting switching behaviour.

**11: R2: Why learning with an uncertainty model helps training and final performance?** See answer 1.

**12: R3: "Dense Human Body" by Wei et al.?** The "Dense Human Body" is concerned with learning descriptors for matching *pairs* of 3D bodies; DensePose learns instead a map from *any single image* to a 3D model, so they solve different problems and their training setup is also quite different (as it is based on a set of classification problems).

**13: R3: Why a Gaussian distribution is a good model?** Because errors usually have unimodal distributions and strong linear correlation, so a Gaussian is a reasonable model.

[Meta-Review · NeurIPS 2019]

This submission received three diverging scores initially 5,5,7. All reviewers pointed out a lack of comparison to related baselines and the authors provided new results in the rebuttal that satisfied the reviewers. One reviewer increased the score, so before AC discussion the scores are 5,6,7. The paper presents a technique for a real world relevant problem. The main concern during the review and discussion phase were. 1. Missing comparison to related work. This is addressed and fixed in the author rebuttal, all reviewers acknowledge the new results and find them sufficient. Please include the results in the final paper. 2. The choice of model components, eg. use of a single Gaussian, prob. U-Net I find this to be a reasonable choice. The statement 13. in the rebuttal states a good fit of the error distribution, but without an empirical result. This could be strengthened by an effort to understand the error distribution better, but it is sufficient for this submission and does not interfere with the claimed contributions. The extension of the prob. U-Net to this problem may be in itself already a contribution. The experiments presented here are much more targeted than those in the U-Net paper. 3. Clarity of the manuscript. This could be improved and I hope the reviews are useful to revise the manuscript. In summary the paper is sufficiently interesting and discusses a real problem for this task of estimating dense poses of humans. The fact that the uncertainty estimates do not improve the results of the method should not withhold the paper but is in the nature of the data. The models could be refined by better understanding of the error distributions and the presented technique describes a possible way to include them. Overall the positive aspects outweigh the negative ones. We hope that the final version will be revised and improved including the results of the rebuttal. Type Table 2 should probably read negative log-likelihood.